# Arctic Futures–Future Arctics?

**Oran R. Young** 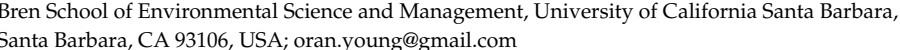

Bren School of Environmental Science and Management, University of California Santa Barbara,
Santa Barbara, CA 93106, USA; oran.young@gmail.com

**Abstract:** Is the Arctic sufficiently distinctive and uniform to justify adopting a holistic perspective in thinking about the future of the region? Or do we need to acknowledge that the Arctic encompasses a number of different subregions whose futures may diverge more or less profoundly? In the aftermath of the Cold War, a view of the Arctic as a distinctive region with a policy agenda of its own arose in many quarters and played a prominent role in shaping initiatives such as the launching of the Arctic Environmental Protection Strategy in 1991 and the creation of the Arctic Council in 1996. Yet not everyone found this perspective persuasive at the time, and more recent developments have raised new questions about the usefulness of this perspective as a basis for thinking about the future of the Arctic. As a result, some observers take the view that we need to think more about future Arctics than about Arctic futures. Yet, today, climate change provides a central thread tying together multiple perspectives on the Arctic. The dramatic onset of climate change has turned the Arctic into the frontline with regard to the challenges of adapting to a changing biophysical setting. Ironically, the impacts of climate change also have increased the accessibility of massive reserves of hydrocarbons located in the Arctic, contributing to a feedback loop accelerating climate change. This means that the future of the Arctic will reflect the interplay between efforts to address the biophysical and socioeconomic consequences of climate change on the one hand and the influence of the driving forces underlying the political economy of energy development on the other.

**Keywords:** adaptation; climate change; energy development; tipping elements

Is the Arctic sufficiently distinctive and uniform to justify adopting a holistic perspective in thinking about the future of the region? Or do we need to recognize that the Arctic encompasses a number of different subregions that may follow different trajectories in the future? Are we concerned, in other words, with Arctic futures or with what may be thought of more accurately as future Arctics? In this reflective essay, I offer some responses to these questions based on my own engagement with Arctic affairs starting in the 1970s. What emerges is a view of the future of the region linked to the consequences of climate change in the high latitudes of the northern hemisphere.

## 1. The Arctic Region

Those of us who developed and promoted the idea of the "Age of the Arctic" during the 1980s grounded our thinking about the future of the northern hemisphere's high latitudes in two major premises [1]. We argued, to begin with, that it was reasonable to approach the Arctic in holistic terms, treating the entire circumpolar North as a spatially delimited area exhibiting enough common features to be treated as an international region in much the same way that we treat Africa, the Middle East, and Southeast Asia as regions. Obviously, such regions are not sealed off from the rest of the world, especially in an era marked by globalization and the rise of digital technologies. However, the prominence of commonalities justifies treating them as regions, especially when we think about issues arising in the realm of public policy that have a distinctive regional character. We asserted, in addition, that while the Arctic had been an important theater of operations for strategic weapons systems during the Cold War, the region was not itself a locus of serious conflicts between or among states. As the Cold War faded, we anticipated, the region would emerge

as an area in which efforts to promote cooperation could unfold somewhat outside the main currents of world affairs.

Taken together, these premises led us to conclude that the Arctic had a policy agenda of its own featuring issues of environmental protection and, more broadly, sustainable development and that we could and should launch cooperative responses to these issues, regardless of the course of tensions or conflicts arising in other regions [2]. For various reasons, this perspective took root in the thinking of members of the policy communities of the Arctic states and especially among leaders in Canada, Finland, and Russia [3]. It evolved into a narrative underpinning the creation of the Arctic Environmental Protection Strategy in 1991 followed by the launching of the Arctic Council in 1996 [4]. Often characterized as the "Arctic zone of peace" narrative, this perspective shaped the practices of the Arctic Council; it continues to inform the efforts of many of those interested in Arctic affairs today.

Yet, others are raising critical questions about the validity of this perspective as a roadmap for the work of the Arctic Council and as a guide to thinking about the future of the Arctic region more generally. The most thoughtful critics are not only questioning the extent to which this narrative is helpful as a framing device in assessing the future of the region; they are also asking searching questions about the extent to which the narrative provided a realistic picture regarding Arctic affairs even in the 1990s. This has brought into focus a consideration of the role of social construction as a central feature of any effort to frame complex issues in ways that make them tractable for consideration in policy arenas and an assessment of alternative perspectives on the past, present, and future of the Arctic in particular [5]. What is the significance of these observations for those contemplating Arctic futures?

## 2. A Uniform Arctic?

Consider first the idea of treating the circumpolar North as an area with enough in common to justify treating it as a distinct region. No doubt, it is correct to say that the Arctic is composed in large measure of remote segments of states whose centers of gravity lie far to the South, that the region is a homeland for numerous groups of Indigenous peoples struggling to maintain the integrity of their cultures and communities in an era of rapid change, and that it is an object of interest to outsiders first and foremost as a source of the raw materials needed to fuel the engines of industrial societies. Nevertheless, differences among the various sectors of the Arctic are also apparent, and these differences have far-reaching implications for how we organize our thinking about the future of the relevant areas.

The Arctic segments of Finland, Norway, and Sweden enjoy a surprisingly moderate climate due to the impact of the North Atlantic Drift. They constitute the homeland of the Sami people, but they have sizable and settled populations of non-Indigenous residents, and they are fully integrated into the social welfare systems of Scandinavia. What is more, these are not recent developments; there have been strong ties between the northern and southern segments of these countries for centuries. As a result, it makes little sense in considering the future of this area to employ the concept of hinterlands as the term is used in discussions of core-periphery relations or internal colonialism. Life in Tromsø does not differ fundamentally from life in Oslo; the same is true with regard to Umeå and Stockholm or Rovaniemi and Helsinki. This accounts for the resistance of many Scandinavians to recent characterizations of the entire Arctic as a coherent and broadly homogeneous region. It explains the development of the idea of the "Old North" as a point of departure for an alternative narrative to employ in efforts to make sense of the past, present, and future of this sector of the Arctic [6].

The North and the Arctic (the two terms are sometimes differentiated in Russian thinking) have loomed large in Russian history and culture for centuries. The associations have not always been happy ones. Convict labor played a major role in Russian (and Soviet) efforts to develop the North. There is a long history of banishing dissidents to remote corners of Siberia; the gulags operating under Arctic conditions in the Russian Far

East were notorious. Nevertheless, the North looms large in most accounts of Russian history. Today, Russia is home to about half of the region's human residents, the majority of whom are settlers in contrast to Indigenous peoples. There is a sense in which Russia is the preeminent Arctic country, and the North remains prominent in Russian thinking, perhaps even more so following the disintegration of the Soviet Union. As a consequence, a number of former Soviet republics with little or no orientation toward the Arctic went their separate ways as independent countries, leaving Russia itself with a more pronounced northern orientation. In rebuilding its economy and reasserting its claim to great-power status, Russia has prioritized the development of the world-class reserves of natural gas in the Arctic, the promotion of the Northern Sea Route as a major artery for commercial shipping, and the strengthening of its armed forces based in the Arctic [7]. Nowhere else does the future of the Arctic figure as prominently in thinking about the future of an entire society as it does in Russia.

The Arctic sectors of Canada and the United States, by contrast, lend themselves more easily to consideration as hinterlands or peripheries. Arctic Canada is sparsely populated, provides a homeland for Aboriginal peoples, and contains raw materials of interest to southern-based corporations. In recent decades, the government of Canada has worked hard to develop enlightened policies in its dealings with the Aboriginal peoples of the Canadian Arctic. Still, the Arctic is a remote place that has little to do with the day-to-day lives of most ordinary Canadians residing in the country's southern reaches, despite their fondness for the idea of the "true North strong and free". For most Americans, Alaska is almost an afterthought. Often derided as "Seward's folly" at the time of its purchase from Russia in 1867 and of interest mainly to those concerned with military security, the extraction of raw materials, and the protection of wilderness areas, the American Arctic does not loom large in the national consciousness. None of this is to deny the legitimate claims of Canada and the United States to be acknowledged as Arctic states. Still, the contrast between the North American Arctic and the thriving social welfare systems of Scandinavia and the central place of the North in the culture and political economy of Russia is striking. Thus, it is not hard to see why many observers are skeptical about the practice of lumping these areas together as components of a relatively homogeneous or uniform Arctic region.

In this account of similarities and differences among the various parts of the Arctic, Greenland stands out as a special case [8]. As a longstanding Danish colony, Greenland has developed a Scandinavian social welfare system along with economic and political systems that are distinctly European in origin. Yet, most Greenlanders are ethnically Inuit and have strong ties to their Inuit brethren spread across the Canadian North, Alaska, and onward to Chukotka. Today, Greenland has developed a thriving political system of its own and is preoccupied with the issue of whether there is a way to develop the economic base needed to support severing its remaining ties with Denmark, establishing itself as an independent, wholly Arctic, and largely Indigenous polity. For its part, Denmark has acknowledged increasingly that its claim to the status of an Arctic state rests with the role of Greenland in Arctic affairs. Denmark announced recently, for example, that Greenland will take the lead henceforth in representing Denmark in the Arctic Council. As the turmoil stirred up by US President Trump's summer 2019 expression of interest in buying Greenland made clear, the future of Greenland itself is linked to shifting relationships among the great powers and the securitization of the general discourse regarding developments in the Arctic. Nevertheless, efforts to come to terms with the issues relating to Greenland's political and legal status dominate the discourse on public affairs within Greenland itself.

## 3. A Peaceful Arctic?

If the premise underlying the holistic view of the circumpolar Arctic as an international region is open to question when we start to look more closely at the circumstances of particular segments of the Arctic, so too is the premise regarding the Arctic as an oasis of peace in a turbulent world. Most disagreements about Arctic matters center on

disputes about the delimitation of jurisdictional boundaries or the interpretation of the provisions of international agreements as applied to specific situations. Canada, Denmark, and Russia have submitted overlapping claims regarding jurisdiction over the seabed extending into the Arctic Ocean beyond the limits of their Exclusive Economic Zones to the UN Commission on the Limits of the Continental Shelf pursuant to the provisions of UNCLOS Article 76. Canada and the United States disagree about the delimitation of their boundary in the Beaufort Sea and about the application of the provisions of UNCLOS Article 37 regarding transit passage to ships plying the waters of the Northwest Passage. A number of signatories to the 1920 Treaty of Paris dealing with the Svalbard Archipelago take the view that Norway's Svalbard Fisheries Protection Zone is incompatible with the provisions of the treaty.

Yet, there is no reason to expect these disagreements to trigger armed clashes in the Arctic. In the 2008 Ilulissat Declaration, the five Arctic coastal states asserted their primacy in matters pertaining to the Arctic and pledged to address any conflicts regarding such matters in a peaceful manner. Canada, Denmark, and Russia have said repeatedly that they will deal with their disagreements about jurisdiction over the seabed of the Arctic Basin peacefully and in accordance with the provisions of UNCLOS Article 76. There is no reason to doubt the sincerity of these statements. Canada and the United States signed an agreement in 1988 agreeing to disagree with regard to their divergent views on the applicability of the provisions of Article 37 to the waters of the Northwest Passage. Nothing has happened that is likely to destabilize this agreement. Norway and Russia were able to negotiate a treaty in 2010 resolving their differences concerning jurisdiction in the Barents Sea and formalizing a suite of cooperative arrangements relating to issues of common interest in this area. No one expects the disagreements regarding the provisions of the Treaty of Paris to erupt into a major conflict over the waters surrounding the Svalbard Archipelago. On the contrary, key players are taking steps to strengthen international cooperation regarding emerging Arctic issues. For example, the 2018 Central Arctic Ocean Fisheries Agreement, a legally binding arrangement committing the five Arctic coastal states and five others to pursuing a precautionary approach to activities likely to affect marine areas lying beyond national jurisdiction, entered into force in June 2021. In contrast to regions such as the Middle East and Southeast Asia, the Arctic seems largely devoid of severe regional conflicts of the sort that could trigger armed clashes.

Still, there is another side to this story that has important implications for the future of the Arctic, especially in the minds of those who see efforts on the part of major actors to maximize relative power and the role of geopolitical factors as the principal drivers of world affairs. To begin with, the use of the Arctic as a theater of operations for strategic weapons systems has never ceased. As Russia seeks to reclaim its status as a great power, it has upgraded the capabilities of the Northern Fleet based on the Kola Peninsula with more sophisticated ships and weapons systems; it has reopened and in some cases expanded military installations abandoned or closed following the collapse of the Soviet Union in the 1990s. Pursuing its objectives mainly through economic initiatives and developing the idea of the Polar Silk Road as a component of its overarching Belt and Road Initiative, China has described itself as a "near-Arctic state" and signaled a clear interest in becoming a significant player in Arctic affairs. The United States has responded to these developments aggressively, deploying the reactivated 2nd Fleet to the Barents Sea, mobilizing war games with an Arctic focus, authorizing the construction of new icebreakers, and initiating plans for upgrading the capacity of its armed forces to operate under Arctic conditions. There are significant disagreements regarding the motivations underlying all these activities. Although no one interprets them as responses to conflicts arising in the Arctic itself, it is reasonable to take note of the increased danger of inadvertent or unintended clashes occurring in the region, an observation underlying the suggestions that there is a need to (re)open lines of communication among leaders of armed forces and to encourage efforts to develop informal codes of conduct governing the deployment of military assets in the Arctic [9].

When it comes to thinking about the future of the Arctic, a disturbing aspect of these developments is the rise of a neo-realist Arctic narrative emphasizing the return of great-power politics to the region and asserting that the Arctic is undergoing a transition form a zone of peace to a zone of conflict [10]. Such views are particularly prominent in the writings of international relations scholars with a newfound interest in the Arctic and the reports of journalists endeavoring to capture the attention of readers with provocative images. However, they also are showing up in the pronouncements of prominent public officials. In a major speech preceding the May 2019 Ministerial Meeting of the Arctic Council, for example, the US Secretary of State asserted that "the region has become an area of global power and competition". He went on to say that, in response, the U.S. is "hosting military exercises, strengthening our force presence, rebuilding our icebreaker fleet, expanding Coast Guard funding, and creating a new senior military post for Arctic affairs" [11].

This great-power politics narrative provides a rationale for advocates of acting vigorously to enhance the capacity to exercise hard power in the region in contrast to the emphasis on softer forms of influence associated with the Arctic Council's vision of the Arctic as an area of unique international cooperation [12–14]. Like all policy narratives, this one features an effort to construct a coherent story in which selected observations about the actions of various players are assembled around a few guiding premises. There are good reasons to adopt a skeptical view regarding the persuasiveness of the great-power politics narrative. However, to the extent that this narrative captures the attention of members of the policy community and of those who shape the content of the public discourse about Arctic affairs, the influence of this Arctic zone of conflict narrative will grow, whether or not the premises on which it is built are persuasive. The result will be a securitization of the discourse regarding Arctic affairs and the development of a perspective on the future of the Arctic that differs sharply from the perspective embedded in the practices of the Arctic Council.

## 4. A "New" Arctic?

Some may draw the inference from this account that we ought to be thinking about future Arctics or the globalization of the Arctic rather than about the future of the Arctic as a distinctive region in our efforts to think carefully about what lies ahead for the lands and waters of the high latitudes of the northern hemisphere and for the peoples who reside there. It is hard to find a central thread linking the concerns of those seeking to come to terms with the economic crisis of the State of Alaska, the complex political future of Greenland, the Atlantification of the Barents Sea, and the extraction of the massive deposits of natural gas in Northwestern Siberia and the adjacent waters of the Kara Sea. At the same time, some see the Arctic as an increasingly prominent arena in the global competition among China, Russia, and the USA. These are all legitimate concerns; they occupy the attention of many who have no particular concern about the consequences of their actions for the future of the Arctic as a distinctive international region. Yet there is another stream of developments now unfolding at an accelerating pace on a circumpolar basis that raise profound questions about the sustainability of both the biophysical systems and the human systems of the whole Arctic and that suggest an alternative perspective on the future of the Arctic. Many associate this perspective with the idea of a "new" Arctic [15].

Taken together, Arctic marine and terrestrial systems constitute ground zero with regard to the onset of climate change [16–19]. Surface temperatures in the Arctic are rising at a rate that has reached three times the rate of change in surface temperatures anywhere else on the planet. Sea ice in the Arctic is receding and thinning at a dramatic pace. Experts now expect that the Arctic Basin will be essentially ice free for some part of the year within two to three decades. Permafrost is thawing at an accelerating rate not only compromising all sorts of infrastructure in the Arctic but also introducing the prospect of large releases of carbon dioxide and methane sequestered in permafrost in the tundra and in methane clathrates in shallow coastal waters in the Arctic. Wildfires are

raging unchecked on an unprecedented scale both in Siberia and in the North American Arctic; some of them burn year around and affect tens of thousands of square kilometers. These developments are changing large ecosystems in ways that are leading to major shifts in the spatial distribution of fish stocks and in the health of populations of marine and terrestrial wildlife. As many observers have concluded, we are now facing a global climate emergency, and the consequences of this looming crisis are nowhere more apparent than they are in the Arctic.

The resultant challenges to the human communities of the Arctic are diverse in some respects. Some coastal communities, for example, are facing an urgent need to relocate as a consequence of severe coastal erosion making their current locations untenable. Others are confronted with growing threats arising from unprecedented flooding and the devastation caused by massive fires. Still others have to find ways to cope with major changes in the abundance and distribution of living resources critical to their livelihoods. From another perspective, however, all Arctic communities face common challenges of adapting to an environment that is not only changing rapidly but also prone to non-linear changes that are difficult to anticipate with any precision. What this means is that the key to the future of human settlements in the Arctic is not an issue of sustaining arrangements that are already in place but rather a matter of introducing dramatic adaptive measures in a timely manner, without compromising the quality of life of their inhabitants [20]. Although the nature of the specific measures required to achieve this goal will differ from place to place, the cultural, economic, political, and social dimensions of adaptation as a societal process have much in common regardless of the distinctive features of the settings in which adaptations to the onset of climate change occur. This suggests that efforts to devise effective adaptation strategies will constitute a central theme with regard to the future of all sectors of the Arctic and that this will provide opportunities to explore the value of cooperative initiatives on a circumpolar basis. At a minimum, those concerned with adaptation to the impacts of climate change in the high latitudes will find it useful to compare notes about the effectiveness of specific responses on a regular basis and perhaps to develop a clearinghouse providing access to information on the results of measures that have been implemented in specific places and to advice for those dealing with similar challenges in other places. The Arctic Council may be able to play a constructive role in this realm [21].

Ironically, the impacts of climate change also have made the Arctic increasingly accessible to those motivated by the economic and political attractions of launching largescale projects aimed at exploiting the natural resources of the region. The most dramatic example so far is the development of the massive deposits of natural gas located in Northwestern Siberia and the adjacent sector of the Kara Sea. Already, large shipments of liquefied natural gas are moving from the new port of Sabetta on the Yamal Peninsula westward to European markets and eastward to Asian markets. And major players, including international investors such as France's TotalEnergies and China's CNPC, as well as the Russian companies Novatek and Gazprom, are taking vigorous steps to accelerate the exploitation of what we have come to realize are massive reserves of natural gas in northwestern Siberia and the adjacent marine areas. Realistic assessments now treat this sector of the Arctic as an area rivaling the Middle East or the Gulf of Mexico as a source of hydrocarbons. Enhanced interest in exploiting the raw materials of the Arctic is apparent in other areas as well, including Alaska, Canada, Norway, and the Russian Far East. A fierce debate over the pros and cons of developing major deposits of rare earths and uranium became the focus of the April 2021 election in Greenland, for example, leading to the fall of the incumbent government and its replacement by a coalition led by the former opposition party.

A striking feature of this development is the strengthening of linkages between the Arctic and the outside world. China has displayed a growing interest in the Arctic as an economic frontier, exploring opportunities to invest in Arctic projects, developing the idea of the Polar Silk Road, and incorporating the Arctic into its globe-spanning Belt and Road Initiative. European and Japanese investors have become prominent stakeholders

in projects focused on hardrock minerals in the Arctic as well as Arctic gas and oil. Korean firms such as DSME have taken the lead in the construction of a new class of LNG tankers that are now delivering natural gas from the Russian Arctic to markets in both Europe and Asia. Commercial shipping in the Arctic is growing accordingly. While this development has focused so far on destinational shipping associated with the transport of natural resources such as natural gas to non-Arctic markets, some major players foresee opportunities arising from the continued impact of climate change to open sea routes in the Arctic linking markets in Asia and Europe.

Of course, Arctic developments constitute only one determinant of the trajectory of both climate change and other change agents operating on a global scale. However, it is hard to miss the disconnect between the spreading crisis of adaptation in the Arctic on the one hand and the sharp rise of energy development and related industrial activities in the Arctic made possible by the impacts of climate change on the other.

## 5. A Paradoxical Future?

With all due respect to differences among the forces shaping the future of specific segments of the Arctic, therefore, there is a sense in which the future of the region as a whole will be determined by how we come to terms with a striking paradox. Nowhere on Earth are the impacts of climate change more dramatic and intensifying more rapidly than they are in the Arctic. As a result, both the biophysical systems and the human systems of the region are experiencing transformative changes, with consequences extending outward to the rest of the Earth system and triggering feedback processes affecting the Arctic. One of these changes centers on the increasing accessibility of the Arctic's massive deposits of hydrocarbons that are attractive to powerful players driven both by economic incentives and by political interests. However, extracting the Arctic's hydrocarbons and delivering them to the industrial societies of the outside world will contribute to sustaining, perhaps even increasing, emissions of the principal greenhouse gases that are the drivers of climate change. A particular concern in this regard stems from the fact that the extraction and shipment of nature gas leads to the release of methane, a dangerous short-lived climate pollutant [22]. The result is the prospect of a future dominated by a powerful feedback loop in which climate change increases the accessibility of hydrocarbons whose exploitation contributes to a continuation and perhaps an acceleration of the pace of climate change. Under the circumstances, familiar narratives such as the Arctic as a zone of peace or as a zone of conflict may become outmoded, overwhelmed by the force of this juggernaut.

Notable in this regard is an apparent inability or at least unwillingness of those who think about the future of the Arctic to come to terms with the tension between the elements of this paradox. There is a sizable community of people who focus on the impacts of climate change in the Arctic and the consequences for the Earth's climate system. There is an equally large community of people who are concerned with the economics and politics of the extraction of the Arctic's energy resources and the options for shipping these resources to outside markets. However, there is a striking disconnect between these communities. There is little overlap in the membership of the two communities; the discourses arising from the deliberations of their members have almost nothing in common, and individual members seldom engage in a focused effort to explore, much less to come to terms with, this paradox. Yet the future of the Arctic may well depend on efforts to address the tension between the increasingly severe biophysical and socioeconomic impacts of climate change on the region on the one hand and the forces driving the political economy of largescale projects focused on extracting and shipping the Arctic's massive reserves of fossil fuels on the other. What is more, the way in which we come to terms with this tension in the high latitudes of the northern hemisphere will have profound consequences not only for the future of the Arctic as a region but also for the future of the Earth system as a whole.

**Funding:** This research received no external funding.

**Institutional Review Board Statement:** Not applicable.

**Informed Consent Statement:** Not applicable.

**Data Availability Statement:** Not applicable.

**Conflicts of Interest:** The author declares no conflict of interest.

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
