# Peer review of "Arctic Futures–Future Arctics?"

_sustainability, doi:10.3390/su13169420_

Round 1

Reviewer 1 Report

This is a really interesting and well-written contribution, perfect for the Special Issue “Shaping Tomorrow’s Arctic.”  The distinction between Arctic Futures and the Future of the Arctic is thought provoking and valuable.  I recommend that the paper be published with minor revisions after addressing these two issues, as well as the minor points raised below.

Line 320: Care is needed with this wording –

But extracting the Arctic’s hydrocarbons and delivering them to the industrial societies of the outside world will fuel continuing emissions of the principal greenhouse gases that are the drivers of climate change.

Conceptually this is true, but over the near term the ice loss in the Arctic will be due to past and present global emissions.  And for the foreseeable future, the emissions from specifically Arctic-generated hydrocarbons are likely to be a small component of future climate change.  A USGS assessment found that of the world’s undiscovered energy resources, the Arctic is projected to hold 13% of its oil, 30% of its natural gas, and 20% of its liquefied natural gas.  More than 80% of these resources are thought to be offshore and will be dangerous to develop for a while, then it will take a while to ramp up production, and by then the global demand for oil and gas may be ramping down.  So in and of itself development of Arctic hydrocarbons is not the main thing driving future opening of the Arctic.

Line 340: Please clarify what you mean in this statement regarding the way in which we address this in the Arctic, and also the profound consequences for the Arctic and the Earth system.

What is more, the way in which we address this contradiction in the Arctic will have profound consequences not only for the future of the Arctic but also for the future of the Earth system as a whole. 

The future of the Earth system as a whole is dependent on

1) how we address climate mitigation, including reducing global emissions, carbon sequestration, etc., most of which will take place outside of the Arctic, and 

2) how the Arctic physical environment responds -- i.e. overall warming/Arctic amplification, glacier melt and sea level rise, disruption of Northern Hemisphere weather, permafrost thaw and methane release, etc.

Minor suggestions, mainly for clarity:

I think that Cold War should be capitalized throughout.

Line 13: maybe add reference to this concept addressed in the text for clarification -- Although the nature of the specific measures required to achieve this goal will differ from place to place, the cultural, economic, political, and social dimensions of adaptation as a societal process have much in common regardless of the distinctive features of the settings in which efforts to adapt to the onset of climate change occur.

Line 18: clarify what you mean by contradiction in this statement “Under the circumstances, the future of the Arctic will depend in considerable measure on efforts to address the contradiction between the biophysical and socioeconomic consequences of climate change and the forces driving the political economy of energy development.” – are you referring to competing interests, mismatched goals?

Line 27:  Young 1985[delete/], 1986

Lines 45-60: for context, maybe bring in the 1961 Antarctic Treaty goal "in the interests of all mankind that Antarctica shall continue forever to be used exclusively for peaceful purposes and shall not become the scene or object of international discord." 

Line 70 and throughout: add a couple more references to publications by others

Line 74: consider inserting a table to highlight/visualize the differences between countries in present/future Arctics

Present/Future Arctics

Core

Iceland, Norway, Sweden, Finland, Russia

From Peripheral towards Core?

Canada

Peripheral

United States

?

Greenland

Line 105: clarify “the [insert -- entire Arctic] region’s

Line 105-109: long complex sentence, break into 2

Line 115: capitalize Aboriginal

Line 146: the future of Greenland itself is linked to broader currents of world affairs – such as? clarify what these broader currents are, provide a couple of examples, the reference to Trump is ambiguous and many may not remember that event, especially in the coming years

Line 149 – note to editor, move subhead to next page for final publication

Line 182: maybe add to this section on Arctic conflicts sanctions imposed on Russia for Ukraine? https://www.nytimes.com/2018/02/28/business/energy-environment/exxon-russia.html 

Line 201: from not form

Line 202: maybe mention the signing of the Ilulissat Declaration in 2008 which was designed to head off the growing calls for a need for some other mechanism of resolving disputes

Line 226: maybe add table to highlight/visualize the trends you’ve identified

Future of the Arctic

Rapid, equitable adaptation --

all Arctic communities face common challenges of adapting to an environment that is not only changing rapidly but also changing in ways that are difficult 267 to anticipate with any precision.

Development – resource exploitation/shipping

Internal interests in development

Increased international interests

Line 225: for clarity maybe remind readers as follows --  differs sharply from the [insert: unique international cooperation”  perspective embedded in the vision statement articulated at the 2013 Ministerial Meeting of the Arctic Council

Line 228: somewhere it would be good to be explicit about what you mean about the “future of the Arctic” vs. “future Arctics” (which were clarified above) and here might be the place, maybe in line 238 “some generic, regionwide sense, as a future of the Arctic”

Line 235: clarify – “they occupy the attention of many who have no particular concern about the 235 implications of their actions for the future of the Arctic as a whole”

Line 239: need something here to establish that you are going to be identifying some common issues in the following. Suggestion: “Yet there is another stream of developments now unfolding at an accelerating pace on a circumpolar basis that raise profound questions about the sustainability of both the biophysical systems and the human systems of the whole Arctic and that have given rise to the idea of a “new” Arctic” – [insert – and are creating common ground, or something like that]

Line 249: replace “melting” with “thawing”

Line 280: replace “clearing house” with “clearinghouse”

Line 311: many others have written about this and should be cited here, here are a few:

Palosaari, T. (2020). Climate change ethics in the arctic. In Climate Change and Arctic Security (pp. 53-60). Palgrave Pivot, Cham.

Newton, R., S. Pfirman, P. Schlosser, B. Tremblay, M. Murray, and R. Pomerance, 2016. White Arctic vs. Blue Arctic: A case study of diverging stakeholder responses to environmental change. Earth’s Future, 4, doi:10.1002/2016EF000356. http://onlinelibrary.wiley.com/doi/10.1002/2016EF000356/full

Petrick, S., Riemann-Campe, K., Hoog, S. et al. Climate change, future Arctic Sea ice, and the competitiveness of European Arctic offshore oil and gas production on world markets. Ambio 46, 410–422 (2017). https://doi.org/10.1007/s13280-017-0957-z

Line 317: with consequences extending [insert: both] outward [insert: and inward] to [insert: /from] the rest of the Earth system.

Author Response

Young, “Arctic Futures – Future Arctics?” Response to Reviews

As always, I have benefitted greatly from the observations of the reviewers of the first draft of this manuscript; I have made substantial changes in the revised version in response to these observations. I have found the critical observations of Reviewer 2 particularly helpful as a stimulus to clarifying my thinking, though I disagree with some of this reviewer’s arguments as I make clear in my specific responses below.

The comments of the reviewers have led me to sharpen the distinction between a reflective essay and a report on research. I have set forth a series of reflections derived from my own experiences as someone engaged in Arctic affairs for almost 50 years. Of course, I want to avoid factual errors and unjustified inferences; I am grateful to the reviewers for helping me to avoid several mistakes. But I regard it as my prerogative to set forth my own perceptions and assessments, even when they diverge from those of the others.

I also feel that an essay of this sort does not require the amount of documentation that would be expected in a report on research. Certainly, some references are useful, and I have added several new references. But I want the reader to engage with the flow of my reflections rather than becoming sidetracked regarding issues of documentation. I also don’t find it particularly helpful to construct summary tables. This, too, seems like a strategy better suited to a report on research than to a reflective essay.

The comments of the reviewers do make me aware of a need to clarify what I mean in using the phrase “Arctic Futures – Future Arctics?” I have added a new introductory paragraph to make the central concern of the essay crystal clear.

I have taken the opportunity of preparing the revised manuscript to adjust the references to the system used by Sustainability.

Reviewer 1

I have clarified and sharpened my observations about Arctic energy development. The USGS estimates are now 15 years old. The proven recoverable reserves of hydrocarbons (especially natural gas) in the Russian Arctic are massive. The exploitation of these reserves has become a cornerstone of the post-Soviet Russian economy. So, yes, it’s important to articulate this argument clearly. But I think my basic point is valid.

My main concern is with future developments in the Arctic. But I have adjusted the text in an effort to clarify linkages between the Arctic and the Earth system.

For reasons set forth in my general comments, I do not see any compelling need to add summary tables to the text.

Otherwise, I have gone through all of Reviewer 1’s specific suggestions and made adjustments to address them. Thank you for directing my attention to these concerns.

Reviewer 2 Report

Review of Arctic Futures – Future Arctics

This reflective essay looks at the Arctic region from various viewpoints including the regional cooperation approach, including Arctic Council, and the musings whether this approach was flawed from the start, given the rise of power politics and “hard security” in the region again in 2020s. Climate change emerges as a transformative force the author discusses for the north with its multiple implications for nature, the economy, transport and climate.

In the following I try to offer some comments which may be helpful in revising the manuscript:

  1. Overall, much of the region’s cooperative history, including the Murmansk 1987 speech, has been said in literature hundreds of times, often by the author himself. Authors like Lassi Heininen have recited the speech and its historical role in opening the Arctic into a peaceful development and cooperation. Either shorten this section, or show something new – for example – what motivated Gorbachev to discuss issues the way he did in 1987? In regionalization of the Arctic, what were the dynamics of creating new political arrangements and structure like the Arctic Council ? Why did the Indigenous peoples were relegated to the status of “Permanen Participants”, ‘lower’ than the SAO – senior officials? This did not solve the Indigenous rights issues or participation, and continues to be a grievance in the Arctic Council.
  2. The discussion of Scandinavian, i.e. Fennoscandian countries and their lack of distinct north-south regions is flawed. Regional analysis shows many metrics to the contrary, i.e. EU looks at these areas as Northern Sparsely Populated Areas – NSPAs, population numbers remain low, Sámi are the distinct Indigenous peoples coupled with the last remaining intact ecosystems of the Nordics located in the Northern parts (southern Sweden and Finland have lost over 95 % of their natural forests for example) and the present climates are distinctly sub-Arctic with temperatures down to -50 C in the northern Nordics. Perhaps northern Norway would be closest to having the infrastructure of the south, but this is mostly the result of the large oil reserves in the country – (another point of analysis missed in the essay of how Norway behaves both in and out of the region and in the Arctic).
  3. Greenland analysis is far too optimistic. Whilst it is true that the political system has evolved for more independence from Denmark and some steps have been taken, the gap between Nuuk (capital) and the regions is massive. Most of the fisheries economy remains in Danish hands, capital jobs in ministries and institutes are serviced by Danish scholars and staff on a 3-4 year rotation who never learn Kalaallit (local Inuit language) and the country still contains massive internal colonial disruption. If the author would like to discuss the “Future Arctics”, we d need to acknowledge the lingering situations of the region’s territories as they are, in line with critical and new scholarship. Ever since 1945 Greenland has been denied rights and reviews under the UN, as per Denmarks joining agreements. The same is in place for Alaska, i.e. when the USA joined the UN, the present and future Indigenous clauses from the UN “where not to touch on Alaska Natives or their land question” (see the joining agreements, footnotes). I.e. the essay could refresh from a deeper power analysis of a regional outlook.
  4. One way of looking at reorganizing some of the essay could happen from the structure-agency lines as proposes above – i.e. who decides, who are the actors and what happens.
  5. Analysis of China, and to some extent Korea, Japan and other Asian powers in the paper is good and welcomed. Arctic Railway process in Norway and Finland by China has not been referred to but remains for that part of the Arctic a significant recent development. Rather surpisingly the author does not discuss the Agreement to Prevent Unregulated High Seas Fisheries in the Central Arctic Ocean as a new example of Arctic governance that included China, Russia, Korea, the EU and others in ways that can be seen as rather progressive. I d welcome a short discussion and an analysis on this topic and its implications for the future of the Arctic.
  6. Author mentions the paradox of how climate change impacts the Arctic in worst ways with cascading implications for the globe and yet Arctic Powers continue to seek natural gas and oil from the new opening regions. This paradox could be articulated in much sharper ways, whilst it is true in the big sense. It s 2021 and not 1995 anymore. The future Arctic/s is a world of multinational corporations, NGOs, tourism vessels, Anthrax being released from the permafrost in Yamal, as well as some of the larger drivers. I would welcome a more nuanced and expanded analysis, especially for example regarding the impact of annexation of Crimea and following sanctions against Russia and how Russia is partly excluded (large US companies cannot deliver parts for the oil and gas development fields), and yet continues a range of environmental cooperation mechanisms for example in the Barents Sea area and through cross-border environmental collaboration with Finland, Sweden and Norway. Whilst space is limited and essay is an essay, I d welcome a more up to date and nuanced view of the present and one way to achieve that is to, for example reduce the historical recounting of 1990s (much of it out there already) and reawakening to the present geopolitical complexity, which could be described as a mix of the old and the new, and the newest, i.e. the unknowns of the region, for which the Agreement to Prevent Unregulated High Seas Fisheries in the Central Arctic Ocean, entered into force on June 2021, is an example of the “new”.

Author Response

Reviewer 2

I am grateful to this reviewer for making me think hard about my take on a variety of topics regarding the Arctic today. I am not persuaded by a number of this reviewer’s comments. But I have tried to sharpen the text to make my views as clear and compelling as possible.

1. I agree that the Arctic’s “cooperative history” has been explored in earlier publications. I have compressed my comments on this period as much as possible. But I do need to include a brief reference to the Arctic narrative of the 1990s as a point of departure for the contrasts to follow.

2. I used to agree with this reviewer’s comments on northern Norway, Sweden, and Finland. But engaging with researchers from this area in recent years has persuaded me to adjust my perspective as described in the text.

3. I’m not sure whether or not my take on Greenland is “optimistic.” But it grows out of interactions with people in Greenland in recent years. I have made some adjustments to clarify this paragraph.

4. The basic structure of the essay is set. It is about the pros and cons of adopting an holistic view of the Arctic as a distinctive region. I appreciate this suggestion, but I am not persuaded that restructuring the essay would be helpful.

5. I have added a brief reference to the CAOFA (abut which I have written extensively elsewhere). The rising interest of non-Arctic states in the affairs of the region is important, and I have tried to make this clear in the revised version of the essay.

6. This comment suggests opening up the analysis to a more general account of recent developments in the Arctic. Of course, these are important, and I have written about them at some length elsewhere. My intention in this modest essay is to stay focused on the theme of “Arctic futures – Future Arctics?” In revising the essay, I have endeavored to maintain this focus, while avoiding oversimplification arising from a failure to recognize some significant developments affecting the Arctic today.

Round 2

Reviewer 2 Report

All answers and corrections have been made.